# UNSUPERVISED-LEARNING OF TIME-VARYING FEATURES

## ABSTRACT

We present an architecture based on the conditional Variational Autoencoder to learn a representation of transformations in time-sequence data. The model is constructed in a way that allows to identify sub-spaces of features indicating changes between frames without learning features that are constant within a time-sequence. Therefore, the approach disentangles content from transformations. Different model-architectures are applied to affine image-transformations on MNIST as well as a car-racing video-game task. Results show that the model discovers relevant parameterizations, however, model architecture has a major impact on the feature-space. It turns out, that there is an advantage of only learning features describing change of state between images, over learning the states of the images at each frame. In this case, we do not only achieve higher accuracy but also more interpretable linear features. Our results also uncover the need for model architectures that combine global transformations with convolutional architectures.

## 1 INTRODUCTION

This paper is concerned with the unsupervised learning of features that predict temporal changes in sequence data. Given a time-varying function $\mathbf{x}(t)$, represented by a sequence of data points $\mathbf{x}_t$, $t = 1, \ldots, N$, our goal is to find a function $\dot{\mathbf{z}}(t) \propto \frac{\partial}{\partial t} \mathbf{z}(t)$ that encodes a representation of the local changes $\dot{\mathbf{x}}(t) \propto \frac{\partial}{\partial t} \mathbf{x}(t)$. In this context, we understand the time-discretized $\dot{\mathbf{z}}_t$ as parameterisation of an (unknown) transformation function $f$ such that $\mathbf{x}_{t+1} = f(\mathbf{x}_t, \dot{\mathbf{z}}_t) = f(\mathbf{x}_t, \mathbf{z}_{t+1} - \mathbf{z}_t)$, where $\mathbf{z}_t$ are unobserved feature-vectors. Recent developments for time-series data focus on finding a representation of the state $\mathbf{z}_t$ and model the time-behaviour $\dot{\mathbf{z}}_t$ implicitly. For example Temporal-Difference Variational Auto-Encoders (Gregor et al., 2019) or Kalman Variational Auto-Encoders (Fraccaro et al., 2017) model the time-behaviour via a transition kernel $p_t(\mathbf{z}_{t+1}|\mathbf{z}_t)$. In these models, the prediction of $\mathbf{x}_{t+1}$ only depends on $\mathbf{z}_{t+1}$ akin to standard Variational Autoencoder (VAE, Kingma & Welling, 2014). In contrast, an explicit representation of $\dot{\mathbf{z}}_t$ offers several advantages:

- Features that are constant within a time-series do not need to be encoded, as $\dot{\mathbf{z}}_t = 0$. Constant features are instead modeled within the transformation function $f(\mathbf{x}_t, \dot{\mathbf{z}}_t)$, which leads to a disentanglement of transformation and content. Moreover, $f$ can be chosen to retain fine-grained structure of $\mathbf{x}_t$, for example by modeling it through diffeomorphic warps of $\mathbf{x}_t$, as is done in deep registration approaches (Yang et al., 2017; Dalca et al., 2018).

- If the feature-space lies on a curved manifold, modeling the tangent-space might be easier than modeling $\mathbf{z}_t$. For example, if the observed transformations are image rotations, the transformation can be represented by elements on the unit circle $\mathbb{S}_1$. While this group can be parameterized with a single variable, the wrap-around occuring between zero and 360 degrees makes it difficult to model it directly. However, if the individual rotations between images are small, $\dot{\mathbf{z}}_t$ can be chosen to encode the tangent-space at $\mathbf{z}_t$ with a single variable.

- Calculating $\dot{\mathbf{z}}_t = g(\mathbf{x}_{t+1}, \mathbf{x}_t)$ can be more precise than modeling it via $\dot{\mathbf{z}}_t = \mathbf{z}_{t+1} - \mathbf{z}_t$. As $\dot{\mathbf{z}}(t)$ is a time-derivative, its precision is highly dependent on noise on $\mathbf{z}(t)$. Thus, computing $\mathbf{z}_t$ and $\mathbf{z}_{t+1}$ independently and computing $\dot{\mathbf{z}}_t$ from those point-estimates will lead to a potentially large error in calculation.

Our approach is similar in spirit to Slow Feature Analysis (SFA, Wiskott & Sejnowski, 2002), which finds signals that vary slowly in time, but does not provide a generative model for $\mathbf{x}_{t+1}$. Further,

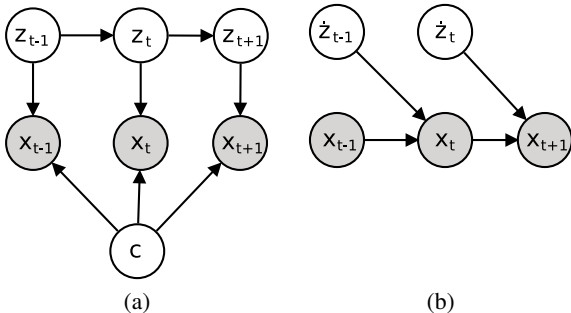

Figure 1: We assume the underlying sequence to follow a graphical model in the form of (a), where $\mathbf{c}$ encodes constant and $\mathbf{z}_t$ encode dynamic features. Observed variables are shaded. Our model uses the graphical model shown in (b) to approximate this sequence without a representation of $\mathbf{c}$.

SFA is known to learn representations for features that are constant over a time-sequence. Because of this, we replace the principle of slowness with the information-bottleneck of the VAE and condition the reconstruction term on previous observations using a conditional VAE (CVAE, Sohn et al., 2015) to remove constant features. This way, our model learns a low-dimensional parameterization of a transformation. This allows us to analyze how commonly used neural network architectures encode geometric information of objects. To our knowledge, our model is the first that learns directly visualizable parameterisations of such geometric features. The focus of the remainder of the paper is to introduce the model and the challenges that it poses for current network architectures. In the experiments, we will train our model on datasets with geometric transformations and analyse how different design choices influence the learned representations.

## 2 TRANSFORMATION ENCODING VARIATIONAL AUTOENCODERS

In this section, we describe the Transformation Encoding Variational Autoencoder (TEVAE). We assume that the generative process underlying the time-series has the form

$$p(\mathbf{x}_1, \ldots, \mathbf{x}_N, \mathbf{z}_1, \ldots, \mathbf{z}_N, \mathbf{c}) = p(\mathbf{z}_1)p(\mathbf{c}) \prod_{t=1}^{N} p(\mathbf{x}_t|\mathbf{c}, \mathbf{z}_t)p(\mathbf{z}_{t+1}|\mathbf{z}_t) \ ,$$

where $\mathbf{c}$ represents a set of hidden features which are constant over the complete time-series and $\mathbf{z}_t$ represent time-varying features. We assume that $p(\mathbf{z}_0) = p(\mathbf{z}_t), t = 1, \ldots, N$, i.e., the process is stationary and $p(\mathbf{z}_{t+1}|\mathbf{z}_t)$ leaves the distribution invariant. We will simplify this model in two ways. First, we will assume that each $\mathbf{x}_t$ carries enough information to identify $\mathbf{z}_t$ and $\mathbf{c}$. Second, we replace $p(\mathbf{z}_{t+1}|\mathbf{z}_t)$ by a random variable that models the difference $\dot{\mathbf{z}}_t = \mathbf{z}_{t+1} - \mathbf{z}_t$. With this, we approximate the generative process as

$$p(\mathbf{x}_1, \ldots, \mathbf{x}_N, \dot{\mathbf{z}}_1, \ldots, \dot{\mathbf{z}}_N) = p(\mathbf{x}_1) \prod_{t=1}^{N} p(\mathbf{x}_{t+1}|\mathbf{x}_t, \dot{\mathbf{z}}_t)p(\dot{\mathbf{z}}_t) \ , \tag{1}$$

This model has the property that the observations $\mathbf{x}_{t+1}$ and $\mathbf{x}_{t-1}$ are conditionally independent given $\mathbf{x}_t$. Thus, instead of modeling complete time-series, we can restrict ourselves to pairs of observations $(\mathbf{x}_{t+1}, \mathbf{x}_t)$. For these pairs, the initial assumption that the time-series must be stationary can be fulfilled simply by adding the time-mirrored pair $(\mathbf{x}_t, \mathbf{x}_{t+1})$ to the dataset. We use this to prevent that a sufficiently powerful $p(\mathbf{x}_{t+1}|\mathbf{x}_t, \dot{\mathbf{z}}_t)$ learns almost deterministic time-transitions without using $\dot{\mathbf{z}}_t$. As $p(\mathbf{x}_{t+1}|\mathbf{x}_t, \dot{\mathbf{z}}_t)$ is typically defined via a deep-neural network, exact inference is not possible and we will use the ELBO instead:

$$\log p(\mathbf{x}_{t+1}|\mathbf{x}_t) \leq \mathbb{E}_{q(\dot{\mathbf{z}}_t|\mathbf{x}_{t+1}, \mathbf{x}_t)} \left\{ \log p(\mathbf{x}_{t+1}|\mathbf{x}_t, \dot{\mathbf{z}}_t) \right\} - \beta \mathrm{KL} \left( q(\dot{\mathbf{z}}_t|\mathbf{x}_{t+1}, \mathbf{x}_t) \| p(\dot{\mathbf{z}}_t) \right) \tag{2}$$

Here, we use $\log p(\mathbf{x}_{t+1}|\mathbf{x}_t, \dot{\mathbf{z}}_t) = -L(\mathbf{x}_{t+1}, f(\mathbf{x}_t, \dot{\mathbf{z}}_t)) + \mathrm{const}$ ,where $L$ is an error function, for example mean-squared-error, and $f$ is the decoder model. We introduced the regularization parameter $\beta$ for notational simplicity and it can be removed by re-scaling $L$. As a prior, we use

$p(\dot{\mathbf{z}}_t) = \mathcal{N}(0, \mathbf{I})$ and we set as encoder $q(\dot{\mathbf{z}}_t | \mathbf{x}_{t+1}, \mathbf{x}_t) = \mathcal{N}(\mu(\mathbf{x}_{t+1}, \mathbf{x}_t), \Sigma(\mathbf{x}_{t+1}, \mathbf{x}_t))$, where $\Sigma(\mathbf{x}_{t+1}, \mathbf{x}_t)$ is a diagonal covariance-matrix. To make sure that the latent space can be interpreted, we add a set of constraints on $q$. First, we enforce that $\dot{\mathbf{z}}_t = 0$ represents the identity. Moreover, if we consider pairs $(\mathbf{x}_{t+1}, \mathbf{x}_t)$ and $(\mathbf{x}_t, \mathbf{x}_{t+1})$, we require that $\mu(\mathbf{x}_{t+1}, \mathbf{x}_t) = -\mu(\mathbf{x}_t, \mathbf{x}_{t+1})$ and $\Sigma(\mathbf{x}_{t+1}, \mathbf{x}_t) = \Sigma(\mathbf{x}_t, \mathbf{x}_{t+1})$. We propose two different models that fulfill these constraints:

- We set $\mu(\mathbf{x}_{t+1}, \mathbf{x}_t) = \mu_{\mathbf{z}}(\mathbf{x}_{t+1}) - \mu_{\mathbf{z}}(\mathbf{x}_t)$ and $\Sigma(\mathbf{x}_{t+1}, \mathbf{x}_t) = \Sigma_{\mathbf{z}}(\mathbf{x}_{t+1}) + \Sigma_{\mathbf{z}}(\mathbf{x}_t)$. This is equivalent to modeling $\dot{\mathbf{z}}_t = \mathbf{z}_{t+1} - \mathbf{z}_t$ with $\mathbf{z}_t = \mathcal{N}(\mu_{\mathbf{z}}(\mathbf{x}_t), \Sigma_{\mathbf{z}}(\mathbf{x}_t))$. It is easy to see that this model fulfills the constraints we impose on the structure through symmetry.

- A relaxation of the first model leads to $\mu(\mathbf{x}_t, \mathbf{x}_{t+1}) = \tilde{\mu}(\mathbf{x}_{t+1}, \mathbf{x}_t) - \tilde{\mu}(\mathbf{x}_t, \mathbf{x}_{t+1})$ and $\Sigma(\mathbf{x}_{t+1}, \mathbf{x}_t) = \tilde{\Sigma}(\mathbf{x}_{t+1}, \mathbf{x}_t) + \tilde{\Sigma}(\mathbf{x}_t, \mathbf{x}_{t+1})$.

In the first model, we obtain an explicit representation of $\mathbf{z}_t$, however, unlike in standard VAE-approaches, there is no structural constraint on it, i.e., we can not assume that $\mathbf{z}$ is normally distributed or lies with high probability in a certain range. For the second model, we do not obtain an explicit representation of $\mathbf{z}$, but allow for non-linear interactions between $\mathbf{x}_t$ and $\mathbf{x}_{t+1}$.

## 3 DECODER ARCHITECTURES FOR THE TEVAE

The fact that the decoder $f(\mathbf{x}, \dot{\mathbf{z}})$ uses a pair of a data-point $\mathbf{x}$ and a feature-vector $\dot{\mathbf{z}}$ makes it difficult to apply standard convolutional architectures on image-data. This is because $\dot{\mathbf{z}}$ might encode a small set of parameters of global transformations, for example an image-rotation, while convolutions apply local transformations to their input. Even though some transformations like image-translations can be reasonably encoded this way — a global translation acts the same on each pixel of the image — most transformations, for example rotations, look very different when applied to different parts of the image. In this case, the global parameter $\dot{\mathbf{z}}$ must be decoded to a local transformation before a convolution is applied. At the same time, the decoder must be powerful enough to ensure that the loss of information in $\mathbf{x}$ is minimized so that a truthful reconstruction of the identity is possible. To our knowledge, no suitable architecture exists that allows for this flexibility on large images. Therefore, for the remainder of this paper we will consider two simple architectures for the decoder of the TEVAE:

**Densely connected layers** Here, $\dot{\mathbf{z}}$ and $\mathbf{x}$ are concatenated as a single input vector to a densely connected feed-forward neural network $f$. This architecture ensures maximal flexibility of the network. However, each layer must have $\mathcal{O}(\dim(\mathbf{x}))$ neurons to ensure that the network can learn the identity function for $\dot{\mathbf{z}} = 0$. Therefore, the number of parameters and computation time grows quickly with the size of the input.

**Image-Registration** If our goal is to model the transformation between images, training a neural network to predict the next image is inefficient. Instead, we can use a neural network to predict an image-transformation, which we can then apply to the image. To do this, let $\mathbf{x} \in \mathbb{R}^{N \times M \times C}$ be an image of size $N \times M$ with $C$ channels, let $\mathbf{p}_{ij} \in [0, 1]^2$ be the position of pixel $\mathbf{x}_{ij}$, $\mathcal{I}_{\mathbf{x}}(\mathbf{p}) \in \mathbb{R}^C$ be the interpolated pixel-value at position $\mathbf{p}$ in image $\mathbf{x}$ and $\mathbf{v}_{ij} \in \mathbb{R}^2$ a translation at $\mathbf{p}_{ij}$. With this, we define

$$\mathrm{Warp}_{ij}(\mathbf{x}, \mathbf{v}) = \mathcal{I}_{\mathbf{x}}(\mathbf{p}_{ij} + \boldsymbol{v}_{ij}) \ .$$

Thus, Warp returns an image where the pixel-values are the interpolated values at the translated image-positions. Finally, we arrive at the definition of the decoder $f(\mathbf{x}, \dot{\mathbf{z}}) = \mathrm{Warp}(\mathbf{x}, g(\dot{\mathbf{z}}))$, where $g : \dot{\mathbf{z}} \mapsto \mathbf{v}$ is a deep neural-network.

## 4 EXPERIMENTS

We conduct two experiments, one on an artifical toy-dataset based on MNIST, the other is based on the CarRacing-v0 task. Note that in the following description, the number of hidden layers and the number of hidden neurons are not tuned. However, we saw qualitatively the same results for different network topologies. For the first experiment, we take images from MNIST and apply image rotations and scalings. In this experiment, the goal is to learn a parametrisation of the applied image transformations, independent of the image content. We create the training dataset as follows: Let $\Phi(\mathbf{x}, \theta, r)$ be an affine transformation of the image $\mathbf{x}$, where $r$ is a scaling parameter and $\theta$ an

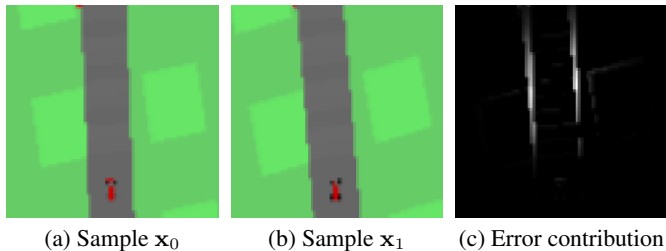

(a) Sample $\mathbf{x}_0$     (b) Sample $\mathbf{x}_1$     (c) Error contribution

Figure 2: A pair of samples $(\mathbf{x}_0, \mathbf{x}_1)$ from the CarRacing-v0 experiment. Shown are $\mathbf{x}_0$, $\mathbf{x}_1$ and the weighted pixel-wise error contribution to $L(\mathbf{x}_0, \mathbf{x}_1)$, brighter indicates larger error-contribution.

angle of rotation around the center of the image. The transformation $\Phi$ warps the pixel positions $\mathbf{p}_{ij} \mapsto \mathbf{m} + r\boldsymbol{R}(\theta)(\mathbf{p}_{ij} - \mathbf{m})$, where $\boldsymbol{R}(\theta)$ is a rotation by angle $\theta$ and $\mathbf{m}$ is the center of the image. The identity transformation is achieved with $r = 1$ and $\theta = 0$. Given an image $\mathbf{x}$ from the MNIST dataset, we create a pair $(\mathbf{x}_1, \mathbf{x}_0)$ using $\mathbf{x}_0 = \Phi(\mathbf{x}, \theta_0, r_0)$ and $\mathbf{x}_1 = \Phi(\mathbf{x}, \theta_1, r_1)$. We sample $r_0$ and $\theta_0$ from distributions $r_0 \sim \mathcal{U}(0.85, 1.15)$ and $\theta_0 \sim \mathcal{U}(-\pi, \pi)$. The first image has therefore a random orientation and a variation in scale. The second image is rotated and scaled relatively to the first by picking $r_1 \sim r_0 + \mathcal{U}(-0.15, 0.15)$. For $\theta_1$ we apply two different ranges: either, we choose $\theta_1 \sim \mathcal{U}(-\pi, \pi)$ and rotate the image arbitrarily relatively to the first, or we pick $\theta_1 \sim \theta_0 + \mathcal{U}(-\pi/4, \pi/4)$, a range between -45 and 45 degrees relatively to the first image. To solve this task, the model has to find a representation of $\dot{\mathbf{z}}$ that encodes the differential transformation $\mathbf{x}_1 = \Phi(\mathbf{x}_0, \theta_1 - \theta_0, r_1/r_0)$.

As model architectures, we combine both encoder approaches with both decoder approaches. We use a 2-dimensional $\dot{\mathbf{z}}$, ELU activations (Clevert et al., 2016) for hidden layers and, if not mentioned otherwise, we use linear activations for output layers. For the encoder architectures, we use a neural network with three dense hidden layers with 98, 12 and 12 hidden-neurons. The output layer encodes mean and log-variance separately for each dimension of $\dot{\mathbf{z}}$. We use this model directly to encode $\mu_{\mathbf{z}}(\mathbf{x})$ and $\log \Sigma_{\mathbf{z}}(\mathbf{x})$, which we refer to as $\mathbf{z}$-encoder. For the second encoder approach, which we refer to as $\dot{\mathbf{z}}$-encoder, that uses $\tilde{\mu}(\mathbf{x}_{t+1}, \mathbf{x}_t)$ and $\log \tilde{\Sigma}(\mathbf{x}_{t+1}, \mathbf{x}_t)$, we just change the input by stacking both input images on top of each other. For the dense-network based decoder (FFNet), we use five hidden layers with $2 \cdot 784$ dimensions each and an output layer with 784 neurons. For the registration-based decoder (Registration), we use five hidden layers in $g$ with 128, 128, 128, 1568 and 1568 neurons. We re-interpret the output of the last hidden layer as $7 \times 7$ image with 32 channels and use a transposed convolution with stride 2, two $3 \times 3$ filters and $\tanh$-activation followed by an image scaling to obtain the final $\mathbf{v}$ of size $28 \times 28 \times 2$. We use border-replication in $\mathcal{I}_{\mathbf{x}}$ to extend the image outside the $[0, 1]^2$ area. For training, we use the mean-squared-error (mean over number of pixels and batch-size) with the Adam optimizer (Kingma & Ba, 2015). We choose $\beta = 10^{-4}$, a learning-rate schedule of $\alpha_t = 10^{-3} \frac{10^4}{10^4+t}$ and a batch-size of 100. We optimize for $T = 10^5$ iterations. In total, the MNIST experiment has 8 combinations of encoder, decoder and dataset. For brevity, we encode this as decoder/encoder/degree, e.g. FFNet/$\mathbf{z}$/$45°$ uses a dense-feed-forward decoder, a $\mathbf{z}$-encoder and the dataset with up to 45 degree rotations between $\mathbf{z}_t$ and $\mathbf{z}_{t+1}$.

For the second experiment, we use the CarRacing-v0 environment of OpenAI-Gym (Brockman et al., 2016) and use the methodology by Ha & Schmidhuber (2018) to obtain a set of 10000 training sequences from random policies of length up to 1000 time-steps. We pre-process the images by applying a $3 \times 3$ Gaussian smoothing kernel and then cutting out an $80 \times 80$ image-region with 3 color-channels excluding the black information-bar of the image. The smoothing removes a few rendering artifacts around edges and corners and allows for the computation of image-gradients as the original dataset only uses flat color-areas. To make the results easier to visualize, we use pairs $(\mathbf{x}_{t+2}, \mathbf{x}_t)$ with a time-stride of two. As in this dataset the car only moves forward between frames, the resulting sequence is non-stationary. We therefore conducted a second experiment where we also add the pairs $(\mathbf{x}_t, \mathbf{x}_{t+2})$.

This task is different from the MNIST task in the way that only parts of the track are visible at any given time. Given the information of the current image, reconstructing the next image in the

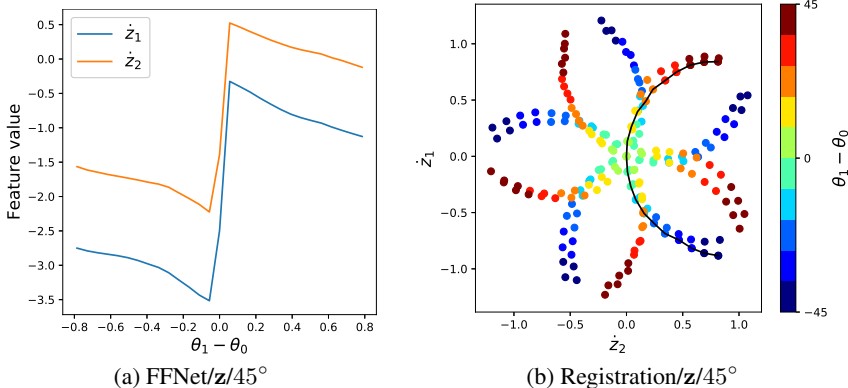

Figure 3: Visualisation of $\mu(\mathbf{x}_{t+1}, \mathbf{x}_t)$ for two of the models depicted in Figure 5 obtained by choosing the same input image and transforming it with $\Phi$, varying $\theta_0$ and $\theta_1$ for fixed $r = 1$. (a): For the model in Figure 5d, fix $\theta_0$ and vary $\theta_1 - \theta_0 \in [-\pi/4, \pi/4]$. It is clear to see that the features approximate a piece-wise linear function with discontinuity. Note that features $\dot{\mathbf{z}}_i < 2$ are outside the visible range of Figure 5d. (b): For the model in Figure 5b, vary $\theta_0 \in \{2\pi \cdot i/10 \mid i = 1, \ldots, 10\}$ and $\theta_1 - \theta_0 \in [-\pi/4, \pi/4]$. Colors encode $\theta_1 - \theta_0$. The black line is an example track of the features for fixed $\theta_0$. Note that $\theta_0$ and $\theta_0 + \pi$ lead to almost the same encoding.

sequence is impossible without additional information. To solve this issue, we use an attention concept to weight the reconstruction of each pixel in the image using the reconstruction error

$$L(\mathbf{x}, \hat{\mathbf{x}}) = \frac{1}{3} \sum_{c=1}^{3} \sum_{i=1}^{80} \sum_{j=1}^{80} w_{ij} |x_{ijc} - \hat{x}_{ijc}| \ ,$$

where $w_{ij} = \frac{1}{Z} \exp((i - 30)^2/20^2 + (j - 40)^2/20^2)$ and $Z$ is chosen such that $\sum_{i,j} w_{ij} = 1$. With this the center of attention is in front of the car, slightly above the middle of the image, while the borders of the image are less important. We show an example of a generated sample-pair and the pixel-wise error contribution to $L$ in Figure 2.

As models, we only use the registration based approach with both types of encoder, as a dense feed-forward decoder would be too large for our GPUs to handle. As before, we use a similar encoder structure for both types of encoders and $\dim(\dot{\mathbf{z}}) = 10$. We use three convolutional layers with 32, 64 and 64 filters of size $3 \times 3$ using a stride of two. Afterwards, we use three hidden layers with 512, 128 and 128 neurons. The output of the last hidden layer is the input of the output layers for mean and log-variance. As before, we use ELU-activations in the hidden layer. The decoder uses the same registration based architecture as in the MNIST experiments with a different number of hidden neurons in the dense layers of $g$. Here we use 4 hidden layers with 10, 10, 800 and 1600 hidden neurons and afterwards interpret the output as image of size $10 \times 10 \times 16$. As before, we use a convolution and image-rescaling to obtain a $\mathbf{v}$ of size $80 \times 80 \times 2$. For training, we use Adam with batch-size of 5, $\beta = 5 \cdot 10^{-5}$ and learning-rate schedule $\alpha_t = 5 \cdot 10^{-5} \frac{10^4}{10^4+t}$ and $T = 5 \cdot 10^4$ training iterations.

## 5 RESULTS

The results of the first experiment can be seen in Figure 5. For each of the eight settings, we overlay a grid of reconstructions of $\mathbf{x}_t$ for varying choices of $\dot{\mathbf{z}}_t$ with a scatter plot showing the means of $q(\dot{\mathbf{z}}_t \mid \mathbf{x}_t, \mathbf{x}_{t+1})$ for varying transformations $\mathbf{x}_{t+1} = \Phi(\mathbf{x}_t, \theta, r)$. For the full image rotation dataset, we show the results of transforming one symmetric and asymmetric number to visualize the differences in encodings. We can see that the FFNet approach learned encodings that form circular shapes, while the registration based approaches learned linear encodings. Especially in Figure 5c we saw a very clear encoding of the first variable as scale, and the second variable as rotation of the image. For the FFNet approaches, we saw a number of artifacts. Even though the models in

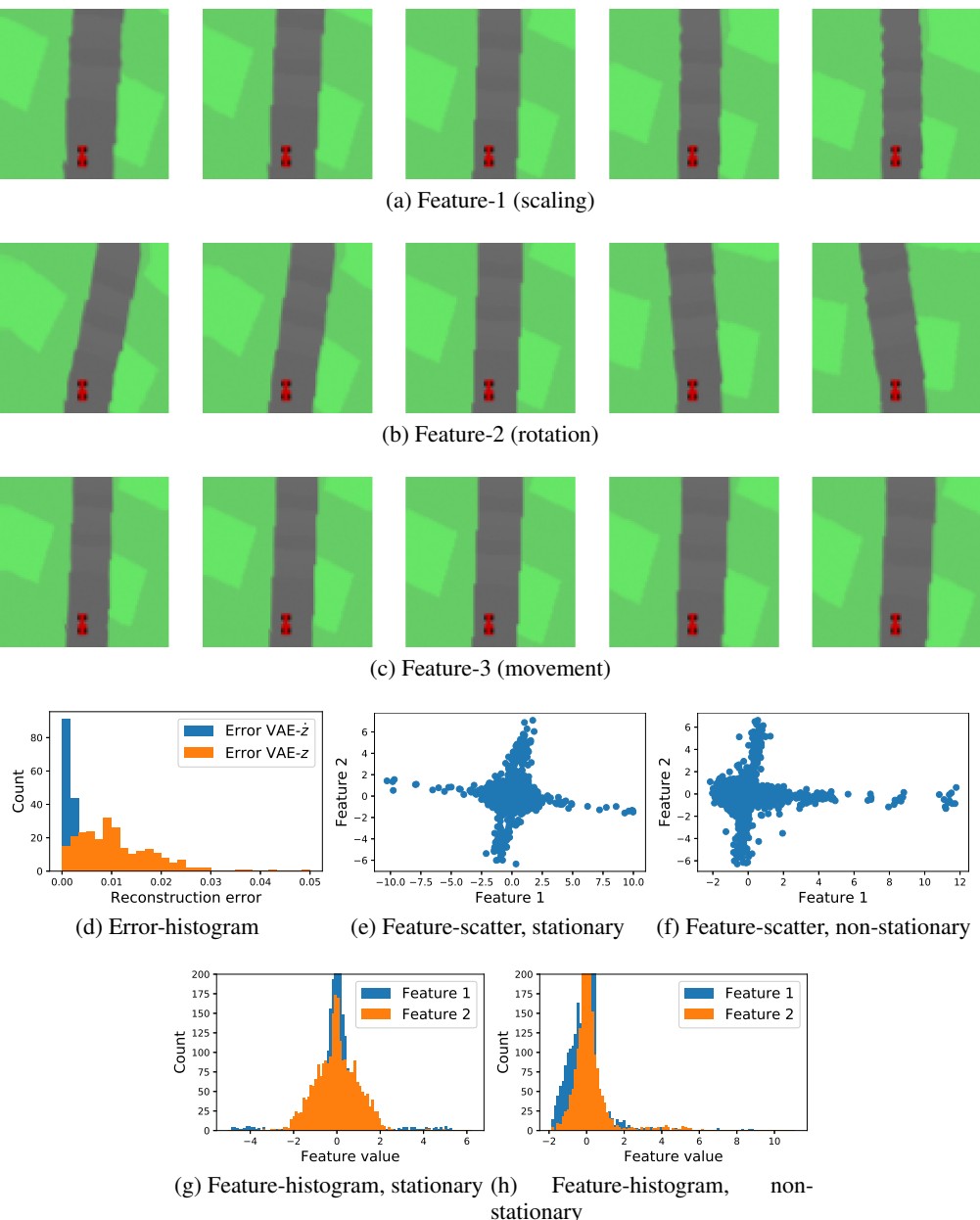

Figure 4: Visualization of the three features with largest variance of $\mu(\mathbf{x}_{t+1}, \mathbf{x}_t)$. (a) – (c): Reconstruction of features with $\dot{\mathbf{z}}$ chosen along the direction of the eigenvectors with largest variance using the VAE-$\dot{\mathbf{z}}$ trained on the stationary dataset. (d): Histograms of reconstruction errors of the the models trained on the stationary dataset. The non-stationary results are similar. (e) – (h): Histogram and scatter-plot of the two features with largest variance in the VAE-$\dot{\mathbf{z}}$-model trained on both datasets. Best viewed on screen.

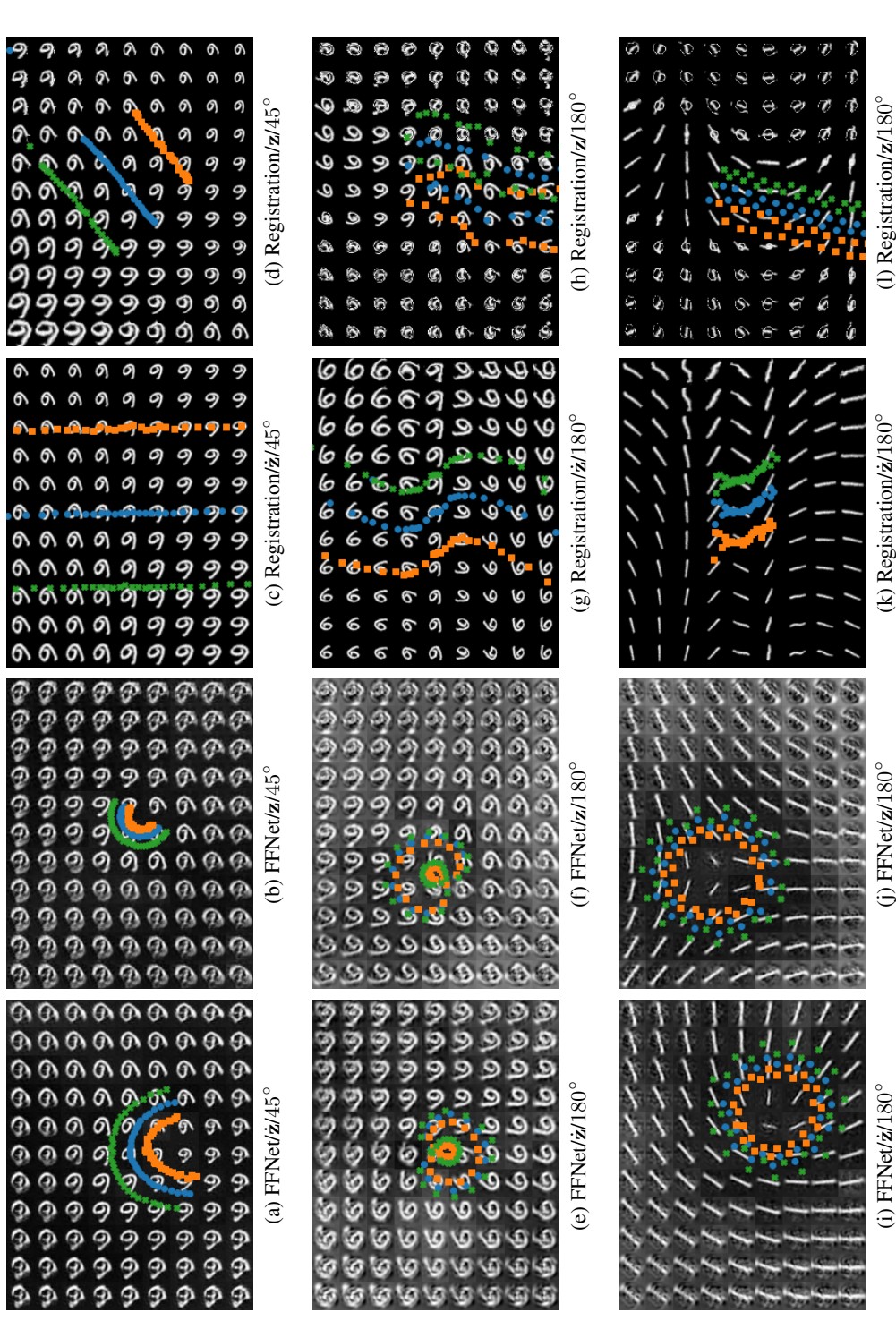

Figure 5: Visualization of the 2D latent space of $\dot{z}$ learned by different architectures on MNIST. Shown are models where the decoder has either an FFNet or registration architecture and the encoder either models $z$ or $\dot{z}$. Models are trained with transformed pairs $(\mathbf{x}_t, \mathbf{x}_{t+1})$ with relative rotations with an angle in either $[-45, 45]$ (top row) or $[-180, 180]$ (bottom rows) degrees. Reconstructions are sampled on a $\dot{z} \in [-1.5, 1.5] \times [-2, 2]$ grid and the reconstruction of the original image is in the center. The overlayed scatter-plot represents the mean $\dot{z}$-values computed by $\mu(\mathbf{x}_{t+1}, \mathbf{x}_t)$ for different transformed versions of $\mathbf{x}_{t+1}$. Colors/marker type symbolize different relative scalings: orange-squares=85%, blue-circle=100% and green-crosses=115%. Each color has 30 points with angles between the min-and max trained angle range. Note that in d), only a fraction of points are visible.

Figures 5a&b were trained with rotations of 90 degrees, the encoding seems to be placed on a half-circle. To analyse this, we took the model from 5b and plotted $\mu(\mathbf{x}_{t+1}, \mathbf{x}_t)$ with $r = 1$ and rotation angles $\theta_0 \in \{2\pi \cdot i/10 \mid i = 1, \ldots, 10\}$ and $\theta_1 - \theta_0 \in \{-\pi/4 + \pi/2 \cdot i/20 \mid i = 0, \ldots, 20\}$. The result can be seen in Figure 3b. We can see that for each choice of $\mathbf{x}_t$, the sampled $\dot{\mathbf{z}}_t$ lie on a half-circle. Moreover, when varying the orientation of $\mathbf{x}_t$, the circle rotates and when we compare the features obtained from $\mathbf{x}_t$ with $\theta_0$ and $\theta_0 + \pi$ we observe that the representations almost overlap. Therefore, by construction of the model, we can conclude that $\mathbf{z}$ lie on a double circle where one rotation is 180 degrees. When training on full rotations, Figures 5e,f,i&j, we see that the rotations of the symmetric number one lie on a full circle, while the asymmetric number six has a double-circle shape, where both circles are connected by a twist, forming a prezel-shape. Closer inspection reveals that each circle maps a range of 180 degree rotations. When we performed similar reconstructions of other numbers (not shown), we saw that the inner circle growed the more symmetric the numbers were until they overlap for fully symmetric numbers. There is no visible difference between the $\mathbf{z}$ and $\dot{\mathbf{z}}$ encodings. For the registration approach, we saw a different set of artifacts. In Figure 5d only a subset of sampled points are visible. A closer inspection of the range of $\dot{\mathbf{z}}$ values, Figure 3b, reveals that the learned encoding of $\dot{\mathbf{z}}$ (and $\mathbf{z}$ by construction) is a piecewise-linear function with large jumps. When plotting a larger area of the space (not shown), we found that the decoder had matching discontinuities that allow for correct reconstruction. Thus, the encoding developed multiple $\dot{\mathbf{z}}$-values that parameterize the same rotation. However, when using the full dataset, the model encoding $\mathbf{z}$, Figures 5h&l only managed to reconstruct one half of the transformations correctly. We did not see any relevant artifacts for the registration model that encoded $\dot{\mathbf{z}}$.

Looking at the second experiment, Figure 4, we observe that encoding $\dot{\mathbf{z}}$ lead to superior performance (Figure 4d). Using PCA, we found that only a 3 dimensional subspace was used, encoding image scaling, rotation and movement. However, the subspace was not axis-aligned with any three $\dot{\mathbf{z}}_i$. In contrast, the model encoding $\mathbf{z}$ failed to model the movement feature. When comparing the stationary with the non-stationary dataset, the models obtained were similar, however with different distribution of features, as can be seen in Figures 4e-h. While all features were non-normal, we see that between the stationary and non-stationary variant the dimension encoding scaling differed a lot. This is because in the task the first few frames are a zoom-in towards the car. We saw a similar effect for the movement direction(not shown), but no effect on rotation, as expected.

## 6 DISCUSSION & CONCLUSION

In this paper we introduced a novel approach to learning transformations in time-sequences. By learning a representation of transformations between images, we show that the VAE can be used to reconstruct and predict image-sequences with high accuracy and find minimal, disentangled parameterisations of the modeled transformations.

Our results allow us to get a glimpse into how neural networks encode geometric features. We show that deep dense neural networks encode image-orientation on a circular structure. In all experiments we obtained two rotations around the origin for encoding the full 360 degrees, with a shape depending on magnitude of transformations in the dataset and symmetry of the image. This circular shape makes it difficult to interpret differences in orientation between images as the direction of the difference vector depends on the position of the orientations on the circle and not only their relative position. We enforced the difference vectors to lie on a flat manifold using a registration based approach where the transformation from difference-vector to warp is independent of the input images. This lead to discontinuous and ambigous encodings as the model tried to flatten the circular structure into a shape where difference vectors form a line. However, when we used a more powerful encoder that did not compute the difference-vector from individual states, but instead from the pair of images directly, we obtained perfectly disentangled, linear features. We hypothesize, this is because we provided the encoder with enough information to locally linearize the features and transform them in a single coordinate-system for the decoder. Our registration based architectures are most similar to Yang et al. (2017) and Dalca et al. (2018) in the deep-registration literature, which can be understood in our framework as using a different type of encoder model. We see the applicability of our approach in the medical domain as well as reinforcement learning to discover and parameterize the space of dynamic image-features without having to learn irrelevant features that do not vary over time. Our results also highlight the need for new architectures that merge global parameterisations with convolutional architectures.

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
