# OpenReview forum: "Unsupervised-Learning of time-varying features"
_ICLR.cc/2020/Conference — Reject_

### Official Review · AnonReviewer1 · 2019-10-07
**Official Blind Review #1**

**Rating:** 1

**Review:**

This paper presents a VAE architecture that separates a fixed content representation and time varying features of an image, that can be used to learn a representation of image transformations in a temporal setting.

The paper is well written and the ideas presented are interesting, but in my opinion not novel enough or thoroughly demonstrated to justify acceptance:

- there is a very relevant work that is not mentioned by the authors and that can be seen as a generalization of the model presented in this paper: "Disentangled Sequential Autoencoder" by Li and Mandt (ICML 2018), which introduces a model that is also disentangling a content and a temporal representation of sequential data. This is basically the more general model introduced by the authors of this submission in the beginning of section 2, without all the assumptions made in the rest of section 2. A comparison with this related work would help assess the differences in terms of modelling power and in performances.

- The assumptions made in this work are fairly strong for most interesting applications, in particular the fact that the content cannot change across time steps.

- To me, the issue with the novelty of this model would not be a big problem if the authors focused more on showing its usefulness in different applications (e.g. medical domain or RL as mentioned in the conclusions). However, the authors only demonstrate the TEVAE on relatively simple experiments that are only tailored to simple image transformations.


**Experience Assessment:**

I have published in this field for several years.

**Review Assessment: Checking Correctness Of Derivations And Theory:**

I carefully checked the derivations and theory.

**Review Assessment: Checking Correctness Of Experiments:**

I assessed the sensibility of the experiments.

**Review Assessment: Thoroughness In Paper Reading:**

I read the paper at least twice and used my best judgement in assessing the paper.

---

> ### Author Response · Authors · 2019-11-06
> **Thanks for the reference**
>
> Thanks for the review and the reference which we will cite and discuss in the paper. Indeed the underlying generative model we start up with is the same as (1) in the reference. However, the main purpose of the paper is that we argue against modeling z_t directly and instead propose our generative model (1) which focuses on the differences between the time-steps. You are right that our model has stronger assumptions insofar that we do not model constant features directly ( c in our paper or f in the reference) and therefore need to assume that c/f and z_t can be correctly estimated from the previous image.
>
> In our paper, we focused on tasks that are simple enough to illustrate the differences in modelling of z_t and \dot{z_t}. We are confident that we could apply it to experiment 4.1 in the reference, which would amount to learning the actions without learning the appearance. However, this task, while looking more fancy, would be easier, because the action-space is discrete(the frame number in the animation) and there is no difficult topology to learn - a flat topology would suffice. In this case, modeling z_t and \dot{z}_t would be equivalent. The experiment 4.3 also sounds very interesting and would require an extension of the model to encode transitions p(\dot{z}_{t+1}|\dot{z}_t, x_t) for the forward-prediction task (to model the elastic collisions). Still, the \dot{z} space would probably be simpler compared to what we consider in our experiments, since movement of the ball is just learning position/velocity, which is a flat space.
>
> In our experiments on MNIST, focusing on scaling and rotation reveals that it is difficult for a VAE to learn an encoding that is compatible with the normal prior and can not be described as disentangled. This is very important, because even if models in Figure 5 e/f/ would produce better visual results, sampling z_t or \dot{z}_t from the prior would not lead to good samples since the true learned encoding lies on a curved manifold. Also, since the manifold changes depending on object symmetry(and involves a twist in 2d space), recent approaches like manifold-based priors are not easy to adapt to this. When enforcing a flat manifold that is consistent with the prior (figure h/l) we obtain that the model fails at reconstructing roughly half of the rotation-space, except when learning a model where the encoder has enough information to perform the parallel transport of the tangent space (Figure g/k). So we claim that the model in Figure g/k is the only one that actually solves the task.
>
> I think the most important result is the carracing task where the z-model fails to learn the translation direction. This is kind of obvious because here it is impossible to disentangle "content" from "position" (without observing x_t there is no valid coordinate system to define position on and movement is the relative shift of all content in the image). The only way to define translation using z_t coordinates would be to treat all "constant/slow changing" features c/f as changing variables z and define translation based on this high level image description of the whole content. This is probably what the encoder in the \dot{z}-model does.
>
> Regarding content: while this is true for the registration model, this is not true for a general NN approach. It is just very difficult to find a proper architecture for that. since changing content is a varying variable, it would be modeled via z_t/\dot{z}_t. E.g. in a video game the only truely constant feature would be "which level am I in".  In our paper, we had to change the loss-function on the carracing task to ignore the borders, otherwise we would have needed a much larger feature space to model all content in the image(or at least the borders).
>
> Final words:
> I agree with your review that our model has its limits. We don't model higher order time-dependencies (which is a simple extension that is only relevant for n-step (n>1) prediction tasks) and we don't model c. This is not a simple extension because (c, \dot{z}_t) is not a complete description of the state and we would also need to model z_t somehow.
>
> In the end, ICLR is about learning representations and we learn a representation for \dot{z} that is meaningful and minimal. One might debate whether this is enough or not, but this might be standpoint specific. We definitely have not answered all questions (e.g. how to get a neural network in the MNIST task to learn the flat manifold as in Figure 5c), but we think there are a lot of tasks out there were our current results are important(e.g. modeling progression of diseases).

---

### Official Review · AnonReviewer3 · 2019-10-22
**Official Blind Review #3**

**Rating:** 1

**Review:**

This paper presents a VAE model. The authors consider time series data and claim that in this situation it is better to model the transformations in the latents instead of each datum independently. The setup is reasonable and seems novel, but since it stops at image registration, which is a well-known existing model I cannot qualify the paper as novel. The paper is mostly clear, some claims are not backed up by experiments and the experiments are lacking. As I motivate below I find the current content more at a workshop level than a conference paper.

Major issues:
* This paper can become a conference paper in two ways in my opinion. 1) It either needs to show that richer modeling has benefits (if anything it would seem from fig 5 that this is not the case). A way towards that would be to take data where there are no simple transformations that we can introduce and show that it discovers reasonable ones. And 2) show on some highly varying temporal domain that this is better than differences of z.
* on page 2 " the initial assumption that the time-series must be stationary can be fulfilled" -- The data doesn't have to comply with our standards of stationarity. A more sensible formulation is we add these additional constraints to our model which are correct if the data is stationary.
* On page 3 "to make sure that the latent space can be interpreted". This is a very strong claim, it implies that if we do this the latent space will always be interpretable, which I think is false and definitely not backed up by experiments.
* The conditions on \dot{z} are interesting and potentially useful and they should be explored in experiments. Putting them in or not does it really make the sense that we think it should make ? Ideally in a setup where the data is not trivial.
* I am not sure what insight a reader can possibly get from figure 3.
* Given the final image-registration setup I find that the following citations are necessary:
jaderberg et al. Spatial transformer networks, Shu et al. Deforming auto-encoders: unsupervised disentangling of shape and appearance.
Minor issues:
* the authors should number all equations.
* In their first equation (not numbered) the indices go beyond N+1.

**Experience Assessment:**

I have published one or two papers in this area.

**Review Assessment: Checking Correctness Of Derivations And Theory:**

I assessed the sensibility of the derivations and theory.

**Review Assessment: Checking Correctness Of Experiments:**

I assessed the sensibility of the experiments.

**Review Assessment: Thoroughness In Paper Reading:**

I read the paper at least twice and used my best judgement in assessing the paper.

---

> ### Author Response · Authors · 2019-11-06
> **some clarifications**
>
> Hi,
>
> Thanks for reviewing our paper!
> * issue 1:
> Figure 5c/g/k (the richer modeling) show an improvement over d/h/l. It is unclear why we don't observe it between 5a/b.  Similarly, in the carracing task, the z-model failed to learn movement direction. see error histogram in 4d that shows a clear improvement of learning the \dot{z} model. Regarding the difficulty of the task, i will copy our answer to another reviewer:
>
> [snip]
> I think the most important result is the carracing task where the z-model fails to learn the translation direction. This is kind of obvious because here it is impossible to disentangle "content" from "position" (without observing x_t there is no valid coordinate system to define position on and translation is the relative shift of all content in the image). The only way to define translation using z_t coordinates would be to treat all "constant/slow changing" features c/f as changing variables z and define translation based on this high level image description of the whole content. This is probably what the encoder in the \dot{z}-model does.
> [/snip]
>
> The carracing task was based on real time-series data and the model found the 3 most important directions. Since carracing is a relevant RL task that got only solved recently, we think that showing that we can discover a parametrization of the movement-space is relevant.
>
> * issue 2: Could you elaborate why this is a major issue? This paragraph states that we can transform a time-series that does not conform to the assumptions into one that does. It is clear that for the "real" time-series only the part of the learned feature-space that actually occurs is relevant, but this procedure ensures that we learn it. Figures 4 e/f/g/h actually show this difference of modeling, which in this task just leads to a symmetrization of the space. There would only be an issue when sampling from the learned distribution as this would only generate stationary time-series. But this is a small issue as we can learn afterwards, which part of the feature-space is relevant.
> * issue 3: We will use a weaker wording here. We mainly wanted to indicate that e.g. length or directions of the \dot{z} vector carry meaning (unlike when, for example, the identity would be placed away from 0).
> * issue 4: Which data would you deem non-trivial? the carracing task is hard to model correctly, even though the underlying transformations turn out to be simple. We are further constrained by the fact that the learned feature-space should still be accessible to human analysis.
> *issue 5: We write for a)  "Figure 3[a] (typo in the paper: we wrote 3b, sorry for this) , reveals
> that the learned encoding of ż (and z by construction) is a piecewise-linear function with large
> jumps. When plotting a larger area of the space (not shown), we found that the decoder had matching
> discontinuities that allow for correct reconstruction. Thus, the encoding developed multiple ż-values
> that parameterize the same rotation." We will clarify the sentence to indicate that we meant a larger space of Figure 5d. The learnt topology in 3a is incompatible with a circle and therefore the decoder has to apply "tricks" to fix the jumps in the encoding. This is the only valid encoding because the model only sees differences of the underlying z-values and for these to make sense, all z-values must lie on a line. If they don't, you get Figure 3b) where we show that the same rotation is encoded differently depending on the orientation of the first image. Therefore the difference-vector \dot{z}_t must be interpreted with respect to z_t. The difference between a) and b) is that the model in Figure 5d) can't do this re-interpretation, while the model in 5b can(Since it observes x_t and thus can compute z_t)
> * Thanks for the references, we will add them to the paper

---

> > ### Comment · AnonReviewer3 · 2019-11-15
> > **thank you for the response**
> >
> > About issue 2: the formulation in the paper was such that it would seem that the data is stationary and you want to add a constraint to the model to make it conform to that. But it's really more like you have some assumptions about invariances that your representation should obey and you enforce with your model. These are not the same things in my mind and I think that is confusing to readers too. The formulation has to change to reflect this.
> >
> > About issue 4: I have never worked on the car racing domain but looking at figure 4 (a, c) are perceptually the same to me. I am not sure what I should understand looking at that figure.
> >
> > About issue 5: anything that is not images I think would be good (eg. sound). The selling point here is that you have an underlying principle to derive better models for temporal data but if that cannot be used to extend the domain of application or improve performance on an existing domain I am not sure about the impact on the audience is.

---

### Official Review · AnonReviewer2 · 2019-10-24
**Official Blind Review #2**

**Rating:** 3

**Review:**

Review of “Unsupervised-Learning of time-varying features”

This work looks at using a conditional VAE-based approach to model transformations of sequential data (transformations can be spatial, such as rotation of MNIST, or temporal, i.e. screenshots of a car racing game). Like how our own visual system encodes differences in time and space [1], they show that in generative modelling with VAEs, “there is an advantage of only learning features describing change of state between images, over learning the states of the images at each frame.”

Such an encoding allows the model to ignore features that are constant over time, and also makes it easier to represent data in a rotating curved manifold. They demonstrate this using data collected from CarRacing-v0 task (a task where agents have to learn to drive from pixel observation of a top-down track), and also on MNIST where the digits are rotated around the center of an image. They provide interesting analysis of the latent space learned and show that indeed this approach can handle both stationary and non-stationary features well (in CarRacing-v0). For MNIST, they compare the latent space learned from transformations (z_dot) and show that this approach can encode image geometric transformations quite well.

While this paper is interesting and highlights advantages of modeling transformations of sequential data, I don't think the contributions are currently sufficient for ICLR conference (right now it is a good workshop paper IMHO). For it to be at the conference level, I can make a few suggestions of things that will bring it there, hopefully the authors can take these points as feedback to help improve the work:

1) Would be great to see how this approach can compare to existing proposed algorithms (i.e. TD-VAE as cited)? Are there problems where this approach will perform really well that current methods are inadequate?

2) As the method is based on an RL-task, would the latent representation learned be useful for an RL agent that relies on the latent code across several representative RL tasks (in both sample efficiency, and/or terminal performance)?

I don't mean to discourage the authors (esp as an Anon Reviewer #2...), as I like the direction of the work, and also appreciate that a lot of effort has gone into this work. I hope to see the authors take the criticism to make their work better. Good luck!

[1] Concetta1988, Feature detection in human vision: A phase-dependent energy model



**Experience Assessment:**

I have read many papers in this area.

**Review Assessment: Checking Correctness Of Derivations And Theory:**

I did not assess the derivations or theory.

**Review Assessment: Checking Correctness Of Experiments:**

I assessed the sensibility of the experiments.

**Review Assessment: Thoroughness In Paper Reading:**

I made a quick assessment of this paper.

---

> ### Author Response · Authors · 2019-11-06
> **Proposal of an RL experiment to improve it**
>
> Hi,
>
> Thanks a lot for your review. Don't worry, reviewer 2 is always appreciated and thought provoking. Also thanks for the reference!
>
> I think you are right that a follow-up experiment would have been very helpful, but we did not want to go overlength and/or cut on the experiment description.
>
> 1) In general, models like TD-VAE would lead to models akin to 5a/b/e/f/i/j insofar that the decoder has to learn to reconstruct the images. If you are not interested in the space of constant-features, this is probably wasteful. In some tasks there might also be no meaningful z-space. For example. if \dot{z} encodes translation, z would encode position. but in carracing, there is no natural coordinate-system to encode position. So you will likely not get a position feature, but instead a set of features that encodes image-content and by comparing the relative position of content, translation can be derived. i.e. we would expect a TD-VAE to learn a much more complicated feature-space for encoding translation. It would probably still perform very good, but the model might not be "physical" or interpretable in any meaningful way.
>
> 2) I think we could use the model in an RL-task. Because when we have a model p(x_{t+1}|x_t, \dot{z}_t) we could obtain a full world-model by learning p(\dot{z}_t|a_t,\dot{z}_{t-1}). where a_t is the position of the agent. And in turn we could use that to implement a Q-function (using Q(x_t,\dot{z}_t)). We felt that this would be a bit too much, because it would move us far from a method-paper to an application domain. But I see this might be required.

---

### Decision · Program_Chairs · 2019-12-19

**Decision:**

Reject

**Comment:**

This work proposes a VAE-based model for learning transformations of sequential data (the main here intuition is to have the model learn changes between frames without learning features that are constant within a time-sequence). All reviewers agreed that this is a very interesting submission, but have all challenged the novelty and rigor of this paper, asking for more experimental evidence supporting the strengths of the model. After having read the paper, I agree with the reviewers and I currently see this one as a weak submission without potentially comparing against other models or showing whether the representations learned from the proposed model lead in downstream improvements in a task that uses this representations.